# Measurement report: Aerosol vertical profiling over the Southern Great Barrier Reef using lidar and MAX-DOAS measurements.

Robert G. Ryan[1], Lilani Toms-Hardman[2], Alexander Smirnov[3], Daniel Harrison[4], Robyn Schofield[1]

[1]School of Geography, Earth and Atmospheric Sciences, The University of Melbourne, Melbourne, Australia.
[2] Department of Defence Graduate Program, Melbourne, Australia.
[3] Science Systems and Applications, Inc., Lanham, Maryland, USA and NASA Goddard Space Flight Center (GSFC), Greenbelt, Maryland, USA
[4]National Marine Science Centre, Southern Cross University, Coffs Harbour, Australia

*Correspondence to*: Robert G. Ryan, rryan1@unimelb.edu.au

For submission to *Atmospheric Chemistry and Physics Measurement Reports*

**Abstract.**

Aerosol vertical profile measurements were made using multi-axis differential optical absorption spectroscopy (MAX-DOAS) and mini-Micropulse LiDAR (MPL) at One Tree Island in the Southern Great Barrier Reef from February to April 2023. This is an understudied location in terms of atmospheric aerosols and chemistry but is growing in importance as multiple research streams examine the influence of aerosols on radiation over the Great Barrier Reef. Solar radiation management proposals, such as marine cloud brightening, require regional-scale aerosol modelling which is evaluated against aerosol extinction and optical depth measurements, necessitating a thorough understanding of measurements of these quantities. MPL aerosol retrieval showed extinction-to-backscatter ratios (0.031 on average) and depolarization ratios (0.015 on average) consistent with clean, unpolluted Southern hemispheric marine aerosol. The maximum depolarization ratio tended to be above the layer of maximum MPL backscatter, consistent with dried sea-salt layers above the boundary layer. MAX-DOAS and MPL extinction profiles show aerosol layers extending beyond 2 km altitude in the morning and to around 1 km in the afternoon. We run the MAX-DOAS retrieval at 360 and 477 nm simultaneously, using the RAPSODI algorithm, finding that this gives much better agreement with the vertically-resolved diurnal cycle of aerosol extinction from the MPL. We also compared aerosol optical depth measurements from integrated MAX-DOAS and MPL extinction profiles, with observations from a hand-held Microtops sun photometer. Mean aerosol optical depth (AOD) values across the campaign compare well, being $0.084 \pm 0.003$ for the Microtops, $0.090 \pm 0.040$ for the MAX-DOAS and $0.104 \pm 0.028$ for the MPL (smoothed to match the MAX-DOAS vertical sensitivity). The diurnal cycles of the smoothed MPL and the MAX-DOAS AOD agreed within uncertainty in most hours of the day, showing a morning peak and afternoon minimum in boundary layer aerosol amount. These measurements show that even in this challenging environment with frequent occurrences of low marine clouds and high humidity, MAX-DOAS (with dual wavelength retrieval) and MPL provide robust methods for probing aerosol vertical distributions and optical depth in the lower troposphere.

## 1 Introduction

Observations of aerosols and aerosol-cloud interactions are crucial to understanding local climate, due to the influence of aerosols and clouds on the Earth's radiation balance (Chen et al., 2021). Aerosols directly scatter incoming solar radiation and indirectly perturb the Earth's energy balance by mediating cloud properties as cloud condensation nuclei (CCN). Understanding the role of aerosol-cloud interactions in local climate has come into recent focus as scientists seek to model the effects of an overall warming world on individual communities and ecosystems and propose aerosol-mediated solar radiation management to protect vulnerable areas (Harrison et al., 2019; Latham et al., 2012).

The Great Barrier Reef (GBR), a spectacular marine ecosystem spanning more than 2000 km in length, is under severe threat due to rising ocean temperatures (Ainsworth et al., 2016). As sea temperatures rise, widespread coral bleaching events are becoming more frequent (Hughes et al., 2017), with widespread coral bleaching occuring at the GBR in 1998, 2002, 2016, 2020, 2022 and 2024 (Readfearn, 2024). The bleaching events can also be influenced by water flow and solar irradiance (Fabricius, 2006).

Marine cloud brightening (MCB) has been proposed as a solar radiation management geoengineering intervention to mitigate coral bleaching at Australia's Great Barrier Reef (Condie et al., 2021; Harrison et al., 2019), under the auspices of the Australian Government's Reef Restoration and Adaptation Program (RRAP). MCB proposes spraying misted sea water into the boundary layer above the reef. This aims to reduce incoming solar radiation through a combination of three effects: (1) direct scattering of radiation by the droplets and liberated sea salt aerosols, (2) sea salt aerosol-mediated enhancement in cloud albedo (Twomey, 1977) and (3) sea salt aerosol-mediated lengthening of cloud lifetime (Albrecht, 1989).

Implementation of MCB relies on a thorough understanding of the natural background aerosols and boundary layer structure at the GBR. Previous aerosol research at the GBR has focused on sources and composition, including sea salt aerosol (Mallet et al., 2016), trace metals (Strzelec et al., 2020), bioaerosols (Archer et al., 2020), continental dust (Chen et al., 2019; Cropp et al., 2013) and secondary aerosol derived from dimethyl sulfide (Fiddes et al., 2022; Swan et al., 2016). Despite this, there is a paucity of information on the vertical distribution of aerosols at the GBR and characterisation of the aerosol optical depth (AOD). Ground-based AOD observations are crucial for validating satellite aerosol and particulate matter measurements, as well as model aerosol column estimates. Australia is currently underrepresented in ground-based AOD measurements compared to much of the rest of the world. The Aerosol Robotic Network (AERONET, aeronet.gsfc.nasa.gov), the foremost global AOD database, has only one site near the GBR (Lucinda, Queensland). Being a coastal site, Lucinda is subject to significant continental influence. AERONET's marine counterpart, the Maritime Aerosol Network (MAN, Smirnov et al. (2009)) has only three datasets in the GBR region prior to RRAP fieldwork campaigns in 2022 and 2023. AERONET data suggests there is a decreasing trend in AOD in the Australian region, as in much of the rest of the world. The 2000-2014 Australian AOD trend is -1.8 ± 3.6 % per year, with the high uncertainty reflecting AOD data scarcity (Mortier et al., 2020).

Light detection and ranging (LiDAR) aerosol vertical profile measurements were carried out at Mission Beach, a coastal site near the northern GBR, in 2016 (Chen et al., 2019). Chen et al., 2019 found the dominant aerosol source to be

continental. To our knowledge, this is the only previous remote sensing of aerosol vertical profiles in the GBR region. Planetary boundary layer heights and dynamics have previously been explored in the Southern GBR using radiosondes (MacKellar et al., 2013; McGowan et al., 2022) and using LiDAR backscatter and drone-based temperature measurements (Ryan et al., 2024). While these studies provide useful constraints on lower atmospheric structure, they do not resolve aerosol layers within or above the boundary layer. Here we present vertical aerosol profile observations in the Southern GBR, using a mini micropulse

LiDAR (MPL) and a Multi-Axis Differential Optical Absorption Spectrometer (MAX-DOAS), throughout January, February and March 2023. Aerosol vertical profiles from these two techniques were integrated to calculate AOD. AOD measurements were also made using direct sun methods and were contributed to the MAN network.

The results in this Measurement Report provide an assessment of the total aerosol amount existing naturally over the Southern GBR, as well as their vertical and temporal variability, during the warmest ocean temperature season, most relevant

to coral bleaching. The vertically resolved aerosol and cloud information provides important context to the proposed MCB experiments by both demonstrating the cloud-altitudes needing to reached by sprayed sea salt aerosols, and demonstrating the natural existence of aerosol layers at that altitude. This measurement campaign provided an opportunity, unique to Australia to date, for the comparison of MAX-DOAS and MPL aerosol profiles. This work also provides an insight into the challenges and optimal analysis methods for interpreting AOD measurements from three different optical techniques, using different types

of remote sensing with measurement viewing geometries.

**2 Methods**

The fieldwork campaign at One Tree Island in the Southern Great Barrier Reef (see Figure 1(a)) ran from 19 January to 7 March 2023. One Tree Island is a small coral island in the south-eastern corner of a shallow coral lagoon (Figure 1(b)), one of several similar, highly biodiverse environments in the Capricorn Bunker Group.

**2.1 Measurement overview**

Atmospheric measurements were conducted at the One Tree Island Research Station, including MPL, MAX-DOAS and a hand-held Microtops sun-photometer. Meteorological information was provided by a weather station at 10 m altitude above the One Tree Island Research Station (Lufft WS800-UMB, https://www.lufft.com/products/compact-weather-sensors-293/ws800-umb-smart-weather-sensor-1790) and a nearby wave-rider buoy (in the ocean outside the reef, location shown in

Figure 1(b)) measuring wave height, wave period and surface wind. Waverider buoy data is publicly available at https://spotters.sofarocean.com/?spotter-filter=SPOT-0311. Cloud information was provided by an all-sky camera (Solmirus All Sky Imaging System M1v, https://solmirus.com/asis-m1v) mounted next to the MAX-DOAS.

The MAX-DOAS instrument deployed at One Tree Island was developed by AirYX GMBH, (https://airyx.de/item/skyspec/). It consisted of a scanner unit mounted on a rooftop facing north-west over the One Tree Island

lagoon. This was connected by fibre optic and data cables to a temperature-controlled spectrometer unit housed inside the

laboratory. The spectrometer unit included UV (wavelength range 300 – 465 nm) and visible (430 – 565 nm) spectrometers, each with 0.6 nm resolution. Only data from the UV range will be discussed in this work. The telescope unit contained an auto-levelled rotating prism for collection of scattered sunlight at specific elevation angles (accuracy < 0.1 °), with narrow field of view (< 0.3º). At One Tree Island, the programmed elevation angles were 1, 2, 3, 5, 10, 20, 45 and 90º. An image of the MAX-DOAS on the rooftop of the One Tree Island Research Station is shown in Figure 1(c).

The MPL deployed at One Tree Island, manufactured by Droplet Measurement Technologies (https://www.dropletmeasurement.com/product/mini-mpl/), uses an eye-safe green (532 nm) laser operating at 2500 Hz. Profiles were collected with 30 s averaging time and a vertical resolution of 30 m. Backscatter profiles were collected in two different polarisation modes (co-polarised and cross-polarised). The MPL at was situated on the beach but well above the waterline in front of the One Tree Island Research Station, as shown in Figure 1(d), with a view to the north-west over the One Tree Island lagoon. Measurements were taken at 0º (horizontal), 2º, 45º and 90º (vertical), however only the vertical measurements will be discussed here. Afterpulse and overlap calibration profiles were recorded several times during the measurement campaign in well mixed atmospheric conditions. The dead time correction supplied by the manufacturer was not altered during the campaign.

Microtops instruments are hand-held sun photometers providing spectral measurements of direct solar radiation. This is used to calculate AOD, aerosol angstrom exponent and total column water vapour at a range of wavelengths. It uses a connected GPS to accurately geo-locate the measurements. In collaboration with the MAN, a Microtops instrument was deployed throughout the One Tree Island campaign. The Microtops used at One Tree Island was manufactured by the Solar Light Company (https://solarlight.com/product/microtops-ii-sunphotometer/). The Marine Aerosol Network Microtops instruments are calibrated at the National Aeronautical and Space Administration's Goddard Space Flight Centre using a reference, stationary sun photometer. Microtops measurements were made at approximately hourly intervals during sunny conditions.

### 2.1 MPL data processing

The mini MPL produces elastic backscatter profiles from a pulsed laser system. The timing of the return signal of a backscattered pulse allows determination of the distance to the scattering object; in vertical, ground-based orientation, this provides information on cloud and aerosol layers. The strategy for retrieving aerosol information from the raw MPL backscatter signals was derived from the seminal work of Welton et al. (2000) and Welton et al. (2002). Firstly, the raw lidar signal needs to be corrected for the afterpulse, overlap and deadtime effects. Secondly, the calibration and range corrected lidar signals are used to calculate the aerosol backscatter-to-extinction ratio. Thirdly, the lidar equation is solved for aerosol extinction in each layer of the MPL vertical profile, with the integrated profiles providing AOD. The MPL used in this study contains information from cross- and co-polarised channels. Following the method in Córdoba-Jabonero et al. (2021), adapted from Flynna et al. (2007), the total raw MPL signal $P$ is comprised of

$$P = P_{co} + 2P_{cross} \tag{1}$$

with linear volume depolarization ratio $\delta_v$:

$$\delta_v = \frac{P_{cross}}{P_{cross} + P_{co}} \tag{2}$$

The raw MPL backscatter signal contains information on Rayleigh scattering from molecules and Mie scattering from aerosols and clouds, background photons at the same wavelength as the MPL and noise from instrumental effects. The raw signal $P$ as a function of range $r$ (range is equal to altitude for the case of vertical measurements) is:

$$P_{(r)} = \left\{ \frac{O(r)EC}{r^2} [\beta_R(r) + \beta_A(r)] \, exp\left[-2\int_0^z \sigma_R(r')dr'\right] exp\left[-2\int_0^z \sigma_A(r')dr'\right] \right\} + N_B + A_{(r)} \tag{3}$$

where $\beta$ indicates backscatter coefficients, $\sigma$ indicates extinction coefficients, and the subscripts $R$ and $A$ indicate Rayleigh (molecular) and aerosol scattering respectively. $E$ is proportional to the laser's output energy and $N_B$ is the background solar radiation contribution at 532 nm. $A_{(r)}$ is the afterpulse calibration function, accounting for the detector being on before each laser pulse is emitted. $O_{(r)}$ is the overlap calibration function, accounting for the difference between the field of view of the output and input lidar signal. The MPL signal is corrected by first subtracting $A_{(r)}$ and $N_B$, then dividing by $O_{(r)}$. $C$ is the instrument-specific proportionality constant between the normalised relative backscatter (NRB) and the MPL signal corrected for background, afterpulse and overlap effects:

$$NRB_{(r)} = C[\beta_R(r) + \beta_A(r)] \, exp\left[-2\int_0^z \sigma_R(r')dr'\right] exp\left[-2\int_0^z \sigma_A(r')dr'\right] \tag{4}$$

An example set of NRB profiles, from 28 February 2023, a mostly cloud-free day at One Tree Island, is shown in (a). The calibration constant should be determined in a region of the atmosphere with as little cloud or aerosol as possible. Welton et al. (2000) recommend a 1 km deep calibration zone which is cloud-free and which has low signal to noise and decreasing NRB with altitude. This allows the assumption that $\beta_R(r) = 0$ at all ranges in the calibration zone (between $r2$ and $r1$). $C_{(r)}$ can then be calculated using tabulated values of $\beta_R(r)$ and $\sigma_R(r)$ from McClatchey (1972), as used in Welton et al. (2000), and constrained using external estimates of the total AOD ($\tau_A$).

$$C_{(r)} = \frac{NRB(r)}{\beta_R(r)} \, exp[2\tau_A] exp\left[-2\int_{r1}^{r2} \sigma_A(r')dr'\right] \tag{5}$$

The values of $\beta_R(r)$ and $\sigma_R(r)$ chosen from McClatchey (1972) were in the 'tropical' category and derived using expected tropical tropospheric water vapour levels and stratospheric ozone levels. We estimated the signal-to-noise ratio (SNR) of the NRB signal by dividing the 10-point moving mean by the 10-point moving standard deviation, as shown for 28 February 2023 in Figure 2(b). SNR calculated in this way showed large variation in the lowest kilometre of the atmosphere due to significant aerosol scattering causing strong backscatter signal fluctuations. In the mid and upper-troposphere, SNR decreases below 10 due to attenuation of the signal in the lower troposphere. Just above the boundary layer, a consistent layer of SNR > 10 was observed, and this region (2-3 km altitude, shown in the black box in Figure 2(b)) was used for calculating the lidar constant $C(r)$. For clear-sky periods on 28 February and 18 March, equation E3 was solved for $C(r)$ in each layer of each vertical profile. We use the aerosol optical depth from the co-located Microtops measurements, taken on the same day, as $\tau_A$ values. The resulting mean and standard deviation of $C(r)$ was $20 \pm 4$. For subsequent aerosol analysis, NRB signals were corrected ($NRB_c$) by dividing by 20.

Aerosol extinction in each layer of each NRB profile is calculated by solving the lidar equation for the aerosol backscatter coefficient $\beta_A(r)$ which is related to the extinction coefficient by the backscatter-extinction ratio: $R_A = \beta_A/\sigma_A$.

$$NRB_c(r) = [\beta_R(r) + \beta_A(r)]\, exp\left[-\frac{2}{R_R}\int_0^z \beta_R(r')dr'\right] exp\left[-\frac{2}{R_A}\int_0^z \beta_A(r')dr'\right] \qquad (6)$$

The Rayleigh backscatter coefficients $\beta_R(r)$ are known (McClatchey, 1972) and the Rayleigh backscatter-extinction ratio $R_R = 8\pi/3$. We apply the iterative strategy described in Welton et al. (2000) to solve for $R_A$, starting from an assumed value of $\beta_A = 0$ at the top of the retrieval grid, and again constraining an aerosol optical depth value from the co-located Microtops measurements. Once $R_A$ is calculated, equation 6 can be solved for $\beta_A(r)$, which is converted to $\sigma_A(r)$ using $R_A$. Finally, vertical $\sigma_A(r)$ profiles are integrated to give aerosol optical depth. We retrieve one value of $R_A$ for the campaign using this method. Retrieved $R_A$ values, and the impact of uncertainty in $R_A$ on the final extinction and optical depth estimates, are discussed in the Results section.

To determine whether backscatter layers in the MPL signal were due to aerosols or clouds, we made use of the raw lidar signals and the co-located all-sky cloud camera. MPL-based cloud detection was necessary because other possible methods of cloud detection at OTI (MAX-DOAS, weather station solar radiation measurements and cloud camera) were only operational during daylight hours. We used the cloud-filtering algorithm developed in Ryan et al, (2024), with validation against the co-located Solmirus all-sky camera, to flag cloudy periods. In Ryan et al, (2024), clouds were identified relevant to the MPL vertical viewing geometry within a 5° radius of the centre of the cloud camera image. For daylight comparison of AOD during the daytime, using the MAX-DOAS and Microtops (which use non-vertical viewing geometries) we also examined the cloud camera images for 'all-sky' cloudiness by using a 50° radius from the centre of the cloud camera image. We also filtered poor or cloudy MPL retrievals by removing outliers with unrealistically large extinction for aerosols (extinction values > 5).

Ryan et al, (2024) also calculated the boundary layer (BLH) from MPL observations, demonstrating good agreement between BLH calculated from MPL NRB and drone-measured potential temperature. The BLH provides important context to this study by indicating the likely extent of turbulent mixing in the lowest part of the troposphere. We use the gradient method, as applied in Ryan et al. (2024) to determine the BLH from each MPL NRB profile: this simple method assigns the BLH as the minimum of the gradient of the MPL backscatter profile.

## 2.2 MAX-DOAS data processing

To derive aerosol information, MAX-DOAS instruments rely on indirect detection through the UV/Visible absorption of the dioxygen complex $O_2$-$O_2$ ($O_4$) (Wagner et al., 2004). This dimer has strong absorption bands at 360 nm and 477 nm which are used in this work to determine aerosol extinction and aerosol optical depth (AOD). The vertical concentration profile of $O_4$ is expected to vary with the square of the atmospheric pressure, meaning that a radiative transfer simulation informed by the atmospheric pressure profile can calculate expected $O_4$ absorptions at each measured elevation angle. Deviations in the

measured $O_4$ absorptions can be attributed to atmospheric scattering, which in a cloudless sky are attributed to the presence of aerosols.

The raw data product from MAX-DOAS instruments is raw solar UV/Vis spectra. From the raw spectra, differential slant column densities (DSCDs) for each trace gas absorbing in the wavelength range of interest are determined by fitting the measured solar spectrum, first corrected for broadband absorbers, with all relevant trace gas cross sections simultaneously (Platt et al., 2008). Here, this is done using the QDOAS algorithm developed at the Royal Belgian Institute for Space Aeronomy (BIRA; QDOAS publicly available at https://uv-vis.aeronomie.be/software/QDOAS/). The wavelength ranges for fitting $O_4$ were 338-370 nm in the UV region and 425-490 nm in the visible region. To cancel out the stratospheric light path, allowing tropospheric-specific gas retrievals, each low elevation angle spectrum was divided by the closest zenith spectrum in time. An example spectral fit for $O_4$ in the UV-wavelength range, from 14:20 on 28/2/2023, is shown in Figure 3(a). $O_4$ DSCD results at 360 nm for a range of elevation angles (not all angles are included for clarity) over the course of 28/2/2023 are shown in Figure 3(b). DSCDs physically represent the concentration ($c_s$) of each trace gas integrated along the tropospheric light path ($L$) of photons reaching the detector for each elevation angle:

$$DSCD = S_s = \int_0^L c_s dl \tag{7}$$

In each layer, the attenuation of the top-of-atmosphere spectrum is termed the slant optical thickness ($\tau(\lambda)$) (Platt et al., 2008; Tirpitz et al., 2022), which is given by:

$$\tau(\lambda) = \sum_s \sigma_s(\lambda) S_s + \sum_i^N b_i(\lambda) + kR(\lambda) \tag{8}$$

Trace gas cross sections are represented by $\sigma_s(\lambda)$, $b_i(\lambda)$ represents a polynomial of degree $i$ (in this work $i = 5$) which approximates broadband absorption features and $kR(\lambda)$ describes the effect of Raman scattering.

A vertical aerosol extinction profile is calculated for each set of elevation angle observations using an inversion algorithm which aims to minimise the difference between the observed and simulated $O_4$ $\tau(\lambda)$ values. Example $O_4$ $\tau(\lambda)$ modelled and measured values from 28/2/2023 at One Tree Island, after the inversion has optimised the modelled values, are shown in Figure 3(c). In this work, we employ the Retrieval of Atmospheric Parameters from Spectroscopic Observations using DOAS Instruments (RAPSODI) inversion algorithm. Retrieval of aerosol profiles from synthetic $O_4$ DSCDs was demonstrated in Tirpitz et al. (2022) and from real MAX-DOAS observations in London by Ryan et al. (2023). In this work, the capability of RAPSODI to retrieve aerosol vertical profiles from multiple $O_4$ wavelengths (i.e. 360 nm and 477 nm) simultaneously is demonstrated for with real DSCD measurements in Australia for the first time. The vertical aerosol profile retrieval strategy is based on the optimal estimation inversion method of Rodgers (2000). To find a solution for a set of atmospheric parameters ($x$) which best reproduce a set of observations, the RAPSODI algorithm minimises a cost function:

$$\chi^2 = (y - F(x))^T S_y^{-1} (y - F(x)) + (x - x_a)^T S_a^{-1} (x - x_a) \tag{9}$$

where $x_a$ is an a priori set of atmospheric parameters, $F(x)$ is a forward model and $S_a$ and $S_y$ describe are the a priori and measurement covariance matrices. The measurement vector $y$ in each retrieval consists of DSCDs at a range of elevation angles (1, 2, 3, 5, 10, 20 and 45°), a particular solar position (characterised by the solar zenith angle and the solar azimuth

angle) and in this case, two wavelengths. RAPSODI uses the VLIDORT radiative transfer model (Spurr, 2008; Spurr, 2006) as the forward model for simulating DSCDs in the inversion. The result of the inversion is the state vector $x$ which contains the amount of aerosols in each layer of the retrieval grid. RAPSODI can be configured to simultaneously retrieve aerosols, from $O_4$ DSCDs, and trace gas vertical profiles such as nitrogen dioxide and formaldehyde, however, here we only report on aerosol retrieval results. The retrieved aerosol concentration profiles can be converted to aerosol extinction profiles, which can

be integrated to produce AOD.

A priori information to initialise the inversion includes an a priori aerosol profile. Here we define this a priori as a profile that decreases exponentially with altitude, characterised by total aerosol optical depth of 0.1 and exponential scale height of 1 km, as shown in Figure 3(d). Figure 3(d) also shows the retrieved aerosol vertical profile matching the modelled-measured comparison in Figure 3(c), showing that the a priori uncertainty (set to 50 % of the a priori profile value in each

235 level) allows the retrieved profile to vary from the a priori. Other inversion input information included surface albedo, taken to be 0.035, a typical ocean value, and temperature and pressure vertical profiles which were taken from the US standard atmosphere adjusted to the surface temperature and pressure at One Tree Island, measured by the co-located weather station. Figure 3(e) shows example averaging kernels for the aerosol retrieval at 14:20 on 28/2/2023. The averaging kernels indicate the sensitivity of the retrieval to the true atmospheric state at each altitude level. Highest sensitivity in the lowest retrieval

layers, and an overall degrees of freedom for signal (trace of the averaging kernel matrix) around 2 are typical of MAX-DOAS results. Retrievals with poor information content or poor convergence were filtered from the final analysis by rejecting results with less than one degree of freedom for signal and with $\chi^2 < 150$.

In order to compare aerosol optical depth measurements from different instruments at different wavelengths, the Angstrom exponent ($\alpha$) can be used. In this case, we take the Microtops AOD at 380 and 500 nm ($\tau_{M,380}$ and $\tau_{M,500}$) to estimate

the ($\alpha$) for converting MAX-DOAS AOD at 360 nm to a comparison wavelength of 500 nm:

$$\alpha = - \left[ \frac{\log\left(\tau_{M,500}/\tau_{M,380}\right)}{\log\left(500\,nm/380\,nm\right)} \right] \tag{10}$$

leading to the calculation of the MAX-DOAS AOD at 500 nm:

$$\tau_{MD,500} \approx \tau_{MD,360}\left(360\,nm/500\,nm\right)^{-\alpha} \tag{11}$$

## 3 Results

The MPL is an active optical method directly probing the atmosphere above the instrument. Because it does not rely on assumptions about light pathlength and horizontal cloud or aerosol homogeneity, it is considered the most robust aerosol and cloud vertical profiling technique for the entire troposphere at One Tree Island. Figure 4(a) shows the time series of vertical NRB profiles, from the surface to 10 km altitude. This figure includes backscatter from clouds and aerosols. Blank spaces indicate data gaps. While there are some occasional instances of strong backscatter towards the top of the troposphere, caused

by free tropospheric cloud, most of the backscatter is recorded in the lowest 3 km of the atmosphere. The mean NRB values

above and below 3 km are 0.112 and 0.294 counts.km$^2$ [($\mu$J) ($\mu$s)]$^{-1}$, respectively. A line of low NRB at around 4 km is an artifact caused by a small spike in the overlap calibration data. The strongest NRB is typically not at the surface, but between 500 and 2000 m altitude, which is expected because of the frequent presence of low marine clouds.

Figure 4(b) shows the time series of depolarization ratio vertical profiles, up to 4 km altitude, encompassing the range of largest NRB values. The depolarization ratio values are typically < 0.05, consistent with an absence of dust or anthropogenic haze (Müller et al., 2007). The depolarization ratio shows a consistent pattern of being < 0.01 under high NRB regions, > 0.02 directly above high NRB regions. The panels of Figure 4 also show the BLH, calculated from the NRB, throughout the campaign. The BLH was typically in the range 0.8-1.2 km throughout (Ryan et al., 2024). In most cases, enhancements in the depolarization ratio were evident immediately above the BLH: the mean depolarization ratio below the BLH was 0.009 ± 0.148, with a high standard deviation due to the occasional presence of clouds, with much larger depolarization ratio, below 1 km. Examining the cloud-filtered aerosol extinction profiles in Figure 4(c), we find the high depolarization ratio values directly above the boundary layer are present during both cloud-flagged and cloud-free periods. This indicates a change in the scattering characteristics (shape and/or composition) of aerosols at or just above the boundary layer, whether clouds are present or not. Elevated layers of high depolarization ratio under marine conditions are consistent with results from Alexander and Protat (2019), who attributed this to the presence of dried sea-salt in disconnected boundary layer sections. High depolarization ratio values are less frequent during the periods when NRB is low (e.g. 9-17 March and 25-30 March). These periods were characterised by thin clouds, rather than being cloud-free, as indicated by comparing the depolarization ratio plot (not cloud-filtered) and the aerosol extinction plot (cloud-filtered).

Highest aerosol extinction values typically correspond to the highest NRB periods, even with the influence of clouds removed (Figure 4(c), e.g. 5-9 and 20-25 March). Early in the campaign, high aerosol extinction values were typically just below the BLH and below the regions of highest depolarization ratio. Around the 19-23 March, high extinction values were recorded above the calculated BLH, although once again the depolarization ratio was above the region of maximum extinction, as shown in Figure 4. Extinction-to-backscatter ratio ($R_A$) values between 0.02 and 0.04 are expected from previous retrievals of in marine environments (Alexander & Protat, 2019; Cattrall et al., 2005; Duflot et al., 2011; He et al., 2006; Omar et al., 2009). At One Tree Island we calculate $R_A$ to be 0.031 ± 0.021 for the whole campaign. This large uncertainty on $R_A$ is considered the largest source of uncertainty in the aerosol extinction calculation. The mean aerosol extinction in the lowest 3 km is 0.021 ± 0.012 km$^{-1}$, using $R_A$ = 0.031. Using the upper and lower bounds of the $R_A$ uncertainty to calculate the mean extinction, we calculate that the uncertainty on aerosol extinction is 18 %. Periods with higher aerosol extinction corresponded to periods with stronger wind speed (Figure 5(a)) from the east-south-east (Figure 5(b)), e.g. 20-25 March. This is likely due to enhanced generation of sea-salt aerosols from sea-spray at high wind speeds. Clouds were more frequent during periods of northerly, low speed wind, e.g. 7-17 March.

In Figure 6 we examine the diurnal and vertical variation of NRB (not cloud-filtered, Figure 6(a)) and aerosol extinction (cloud-filtered, Figure 6(b)) measured by the MPL at One Tree Island. Figure 6(a) confirms that the highest NRB values are predominantly below 1.5 km altitude and mostly at or below the boundary layer height. NRB at low altitude is

slightly higher during the daytime than overnight. After 3 pm local time, NRB enhancements are observed in the free troposphere, growing in altitude from around 6 km to 8 km as the afternoon progresses. This is likely due to the afternoon convective development of free tropospheric clouds. Aerosol extinction is higher and more variable in the daytime (mean $0.018 \pm 0.035$ km$^{-1}$ between 8 am and 6 pm) than at night (mean $0.011 \pm 0.015$ km$^{-1}$ otherwise), which may indicate that evaporative, temperature-driven or photolytic processes are responsible for aerosol formation. Higher aerosol extinction in the daytime may also be driving the slightly higher daytime NRB. Mean daytime aerosol extinction vertical profiles reveal that aerosol layers with extinction > 0.05 km$^{-1}$ and even > 0.15 km$^{-1}$ often extend beyond 2 km altitude. The vertical extent and magnitude of high aerosol extinction levels is greatest throughout the morning, before decreasing again in the afternoon. Ryan et al. (2024) found that the planetary boundary layer height measured using MPL backscatter, and drone-based temperature measurements, showed very little diurnal variation (also shown in Figure 6(a)). That paper showed that the boundary layer height was typically around 800 m altitude in the middle of the day, suggesting that the aerosol layers detected up to 2 km were extending beyond the boundary layer. The elevated aerosol extinction above the boundary layer in the GBR environment is most likely transported from further afield or from evaporation from cloud tops (Braga et al., 2025).

The diurnal mean maximum aerosol extinction measured by the MAX-DOAS (Figure 6(c)) occurs in elevated layers between 0 and 2 km altitude in the morning and around 0-1 km altitude in the afternoon. This is consistent with the presence of aerosol layers above the boundary layer in the morning shown in the MPL aerosol extinction plot. MAX-DOAS aerosol extinction is close to zero above 3 km altitude, however the MAX-DOAS averaging kernels indicate very low sensitivity in the upper retrieval layers (e.g. Figure 3(e)). The mean MAX-DOAS aerosol extinction in the lowest 3 km is $0.038 \pm 0.079$ km$^{-1}$. Note that the while the magnitude of the MAX-DOAS and MPL aerosol extinction values appear similar, they are not directly quantitatively comparable because the MAX-DOAS extinction reported here is at 360 nm, the MPL at 532 nm.

The vertical sensitivity of the MAX-DOAS observations is strongly limited above 2 km, as shown in the averaging kernels in Figure 3(e) and the vertical resolution is also coarser than that of the MPL. As a result, to achieve a like-for-like MPL-MAX-DOAS comparison, we plot MPL results convoluted to the vertical resolution of the MAX-DOAS, and smoothed using the MAX-DOAS averaging kernels, in Figure 6(d). To account for the fact that the lidar profile's lowest retrieval altitude is 120 m, due to limited overlap between the optics' field of view and the return signal, the MPL signal below 120 m is set to the value at 120 m. This follows the ceilometer-MAX-DOAS comparison strategy in Frieß et al. (2016). The result can be thought of as the profile the lidar would retrieve if it had the vertical resolution and sensitivity of the MAX-DOAS. Compared to the unaltered MPL results in Figure 6(b), the smoothed MPL aerosol layers do not extend as high in altitude, due to the lack of vertical sensitivity. This suggests that the MPL is a better technique than the MAX-DOAS to track the evolution of elevated aerosol layer above 2 km altitude. The altitude of the aerosol layer in the smoothed MPL case is also slightly lower than for the MAX-DOAS. However, the diurnal variation is qualitatively very similar between the smoothed MPL and MAX-DOAS results, with both showing greatest lower tropospheric aerosol extinction (at 0-2 km for MAX-DOAS, 0-1.5 km for the MPL) between 8 and 10 am local time.

Finally, we compare aerosol optical depth (AOD) results from the MPL, MAX-DOAS and the Microtops sun photometer in Figure 7. Because there were few cloud-free times in the campaign with all three instruments sampling simultaneously, we present the diurnal mean profile from each method rather than the campaign timeseries. Microtops AOD results are reported as calculated by the instrument, at 500 nm, the closest wavelength to the MPL wavelength (532 nm). Raw MPL AOD values are calculated at 532 nm by integrating the retrieved extinction over the entire MPL tropospheric altitude range (120 m to 10 km). Smoothed MPL results are calculated by integrating the extinction smoothed using MAX-DOAS averaging kernels, at the vertical resolution of the MAX-DOAS as described above, between 0-5 km altitude. MAX-DOAS retrieved extinction is integrated over 0-5 km altitude, for direct comparison to the smoothed MPL AOD, and because there is no sensitivity for the MAX-DOAS (averaging kernels very close to zero) above this altitude. In addition, the MAX-DOAS raw AOD at 360 nm ($\tau_{MD,360}$, calculated either using a single or dual wavelength retrieval) is converted to a 500 nm AOD value ($\tau_{MD,500}$) using the Angstrom exponent as outlined in Equations 10 and 11.

The diurnal cycle of AOD is shown in Figure 7(a) and (b) and the diurnal cycle of $\alpha$ is shown in Figure 7(c). AOD calculated using Microtops, MAX-DOAS and smoothed MPL AOD results are only reported during daylight hours. MAX-DOAS AOD using a single wavelength retrieval (360 nm) is in Figure 7(a), dual wavelength retrieval results are in Figure 7(b). In addition, the averaging kernels used to smooth the MPL results are from the single wavelength MAX-DOAS retrieval in Figure 7(a) and the dual wavelength retrieval in Figure 7(b). The mean AOD values at ≈500 nm from all instruments across the campaign in Figure 7(a) are: Microtops 0.083 ± 0.002, MAX-DOAS 0.090 ± 0.032 (single wavelength) and 0.089 ± 0.040 (dual wavelength retrieval), raw MPL 0.101 ± 0.028 and smoothed MPL 0.104 ± 0.028. In the dual MAX-DOAS retrieval case, Figure 7(b), the smoothed MPL mean changes to 0.091± 0.028 and the MAX-DOAS mean hardly changes, to 0.090 ± 0.040. Despite the similarity of mean AOD values when moving to the dual wavelength retrieval, compared to the single wavelength results, the diurnal cycles alter markedly. Figure 7(b) shows agreement between the smoothed MPL and MAX-DOAS diurnal cycle with a morning peak followed by mid afternoon minimum, while the raw MPL and Microtops diurnal cycle tends toward a midday peak. We found no difference in the final AOD diurnal cycles for MAX-DOAS or MPL when considering a wider cloud-filtering threshold from the co-located cloud camera. It made no difference if we applied the cloud filter to only the centre of the cloud camera image, or also invoked a cloud filter from the entire image. This suggests that differences in the AOD amounts between instruments is driven more by differences in vertical structure and sensitivity than horizontal inhomogeneity.

To place these AOD results in context, we examined previous Microtops measurements taken in the Southern Pacific Ocean region close to Australia, that are reported in the MAN database. The relevant voyages are the *RV Alis* (2016), *RV L'Atlante* (2015) and *RV Melville* (2009-10). The mean AOD from these voyages was lower than any of the means at One Tree Island, at 0.063 ± 0.027. A higher value is to be expected at One Tree Island as it is closer to continental Australia than any of the voyages contributing to the MAN mean. The fact that there are only three previous MAN datasets on AOD in the vicinity of the GBR and north-eastern Australia emphasises the importance of collecting and reporting aerosol information in this region.

The observed AOD over OTI is placed in the context of publicly available AOD information, by comparison to forecasts from the Copernicus Atmospheric Modelling Service (CAMS). Forecasts for AOD and a range of other atmospheric chemistry, aerosol and radiation variables are available at https://ads.atmosphere.copernicus.eu/ (Peuch et al., 2022) and are integrated into numerous widely accessed weather and air quality online platforms. The campaign mean archived forecast AOD value, available at 10 am local time for OTI, is $0.167 \pm 0.069$, shown in Figure 7(a). The much higher campaign mean for CAMS is partly because the timing is close to the diurnal cycle peak time for the MAX-DOAS and smoothed MPL in the dual wavelength retrieval case, and at this time the model value is within uncertainty of the observations. The high variability of forecast AOD over the campaign, quantified by a standard deviation of 0.069, is due to only a handful of days where CAMS AOD exceeds all the measured values by a factor > 2.5.

Error bars in Figure 7 are the standard deviation of the hourly mean values. The raw MPL AOD has much larger variability than the other techniques. This variability, caused by frequent occurrence of high extinction values, likely indicates that the cloud-filtering algorithm is missing some optically thin cloud above the boundary layer. This results in misinterpreted aerosol information both due to enhanced lidar return and because the extinction-to-scatter ratio calculated for aerosols, $R_A$, will be inappropriate for thin cloud. This interpretation is supported by the agreement in magnitude and diurnal cycle shape of the MAX-DOAS and smoothed MPL in the dual wavelength retrieval case. This shows that in a like-for-like comparison in terms of vertical sensitivity, which is only in the lowest km for the MAX-DOAS, both techniques agree well. The raw MPL daytime diurnal cycle is significantly enhanced by being able to observe with higher sensitivity, regions above the boundary layer. The diurnal cycle of smoothed MPL and MAX-DOAS also compares well with the Microtops in the afternoon, although not the early morning, because of more morning cloudy periods.

## 4 Discussion and Conclusions

In this paper we present 5 weeks of aerosol vertical profile measurements at One Tree Island in the Southern GBR, using MAX-DOAS and mini MPL measurements. To our knowledge these results are the first set of remote-sensed aerosol profiles measured entirely in the marine environment of the GBR to be reported in the literature. We also report on the column aerosol amounts from the MAX-DOAS and MPL methods, alongside reference Microtops observations contributed to the Marine Aerosol Network and archived, publicly available forecast data.

Aerosol retrieval using the MPL yielded extinction-to-backscatter ratios consistent with marine aerosol (mean $R_A$ across the campaign $0.031 \pm 0.021$). Using the polarisation capability of the MPL, we calculated depolarization ratios around 0.015, typical of small aerosols in an unpolluted, marine environment (Müller et al., 2007). Higher depolarization ratios tended to be above the layer of maximum MPL backscatter and aerosol extinction, consistent with the presence of dried sea-salt layers above the boundary layer (Alexander & Protat, 2019). Aerosol extinction profiles retrieved using the MAX-DOAS and the MPL both show aerosol layers extending beyond 2 km altitude in the middle of the day, which is beyond the planetary boundary layer heights reported in Ryan et al. (2024) and also shown here. The elevated aerosol layers observed here using remote

sensing methods are consistent with the in-situ observations in Braga et al. (2025), where elevated aerosol layers above cloud top heights were seen away from clouds and attributed to cloud processing.

The ability to measure aerosol column amounts, layer heights and vertical profiles is also crucial in the context of marine cloud brightening experiments at the GBR. Significant work is already underway to better characterise aerosol amounts and composition in the boundary layer above the GBR using in-situ methods (Braga et al., 2025; Hernandez-Jaramillo et al.,

2024). For example, Hernandez-Jaramillo et al. (2024) describe an airborne research facility suited to short-term observation campaigns. While such platforms provide high temporal and spatial resolution within the campaign, a significant advantage of remote sensing platforms such as MAX-DOAS and MPL is that they can be deployed for longer time periods. The agreement in AOD amounts and diurnal cycles between MAX-DOAS (dual wavelength retrieval) and MPL shows that both techniques are suitable for probing boundary layer aerosol distributions. We therefore suggest that deployment of one or both techniques

to a location like One Tree Island, over a long time period, would be advantageous to determine long-term and season boundary layer aerosol variations at the GBR. However, our results highlight two important points about deploying these different sensors for long-term applications. Firstly, the low sensitivity of MAX-DOAS above the boundary layer means that technique misses a significant proportion of the aerosol extinction column in this environment. Secondly, running the MAX-DOAS retrieval at 360 and 477 nm simultaneously is a more robust method of examining boundary layer aerosol extinction and optical

depth, compared to at 360 nm only.

The importance of examining AOD is highlighted by the range of possible AOD sources. AOD is a metric for considering aerosol/radiation metrics available from satellite platforms, ground-based remote sensing and modelling frameworks (e.g. CAMS), in cases where vertically-resolved measurements are not available. Therefore, building trust in available AOD measurements is crucial. The comparison of AOD measurements between instruments at One Tree Island found

mean values of $0.084 \pm 0.003$ for the Microtops, $0.090 \pm 0.040$ for the MAX-DOAS (dual wavelength retrieval) and $0.091 \pm 0.025$ for the MPL (smoothed for comparison with the MAX-DOAS). Good diurnal cycle agreement was found between the MAX-DOAS and smoothed MPL diurnal cycle and also with the afternoon part of the Microtops diurnal variation. The morning value for these instruments also agreed within uncertainty with the forecast AOD from the Copernicus Atmospheric Modelling System, a widely accessed source for public information on AOD, providing confidence in our understanding of

the daily variability in boundary layer AOD levels. In future work it would be good to examine the reasons for the morning peak in lower tropospheric aerosol extinction and optical depth.

The challenge in interpreting AOD observations from different platforms, with different measurement geometries and spatial footprints, was noted in Omar et al. (2013), who compared AERONET observations to satellite lidar observations made with the Cloud-Aerosol Lidar with Orthogonal Polarization (CALIOP) instrument on the Cloud Aerosol Lidar Infrared

Pathfinder Satellite Observations (CALIPSO) satellite. The results presented here in our work provide insight into the challenges of comparing vertical column aerosol measurements from three different optical techniques. The measurement geometries are very different: Microtops looking directly at the sun, lidar probing vertically and the MAX-DOAS collecting photons over a wide vertical and horizontal footprint. The different viewing geometries also mean that, in a landscape such as

One Tree Island with frequent scattered cloud at a wide variety of altitudes, 'clear-sky' conditions are rarely the same for each instrument. It also means that if aerosols are not horizontally homogenous, very different aerosol extinction and AOD results can be expected. The depolarization ratio results presented here support the existence of dried sea-salt particles above clouds. This, along with the ubiquity of low marine clouds that are moving and evolving rapidly, including evaporating, means that aerosol horizontal homogeneity may be an inappropriate assumption. Horizontal aerosol gradients in this environment could also be introduced by different aerosol production mechanisms between the reef and the open ocean, for example due to wave action at the reef edge. Several studies have also explored the potential for individual reefs in the GBR to produce secondary aerosol from dimethyl sulfide emissions (Fiddes et al., 2022; Jackson et al., 2018; Jones et al., 2018; Modini et al., 2009; Swan et al., 2016), a process which could lead to substantial aerosol spatial inhomogeneity. Horizontal variations in boundary layer structure associated with reefs (MacKellar et al., 2013; McGowan et al., 2022) may also be a factor contributing to a highly variable spatial aerosol landscape, contributing to AOD discrepancies between measurement techniques. For example, the MAX-DOAS footprint involves photons traversing more or less of the reef and lagoon environment at different times of day. The viewing direction of the instrument, as shown in Figure 1, would lead to a longer light path-length over the reef and lagoon in the afternoon, when the sun is in the western sky, and a longer light path over the open ocean in the morning. This effect could compound the impact of horizontal atmospheric inhomogeneity on AOD differences between the MAX-DOAS, MPL and Microtops. Nevertheless, in this work we demonstrate that good agreement can be found in the AOD and aerosol extinction diurnal cycles between active and passive remote sensing even in this challenging environment, so long as wavelength-dependent information is provided to the MAX-DOAS retrieval, and the differing vertical sensitivities are appropriately accounted for.

**Competing interests**

The contact author has declared that none of the authors has any competing interests.

**Data Availability**

The data used in this manuscript is publicly available at doi.org/10.26188/25868881 (Ryan & Schofield, 2023).

**Acknowledgments**

The Reef Restoration and Adaptation Program is funded by the partnership between the Australian Governments Reef Trust and the Great Barrier Reef Foundation. The authors thank Heinrich Bruer and Ruby Holmes at the One Tree Island Research station for their assistance during the field campaign, as well as Lauren Hasson and Adrian Doss for fieldwork assistance and help with accessing all-sky camera images.

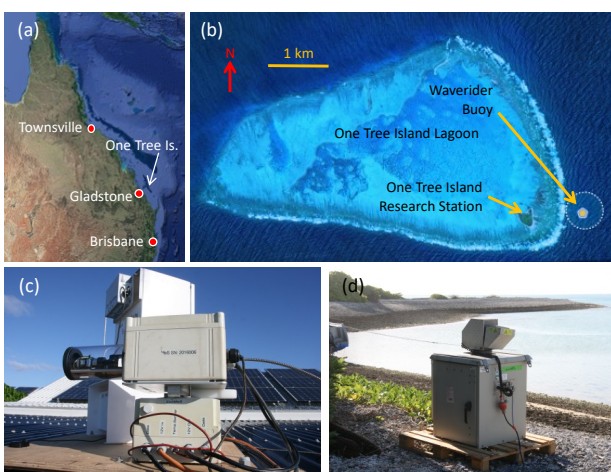

Figure 1: (a) Map of Queensland showing the location of OTI, in the Capricorn Bunker Group off the coast of Gladstone. (b) Satellite image showing the OTI lagoon and the island itself, as well as the location of the waverider buoy outside the lagoon. (c) MAX-DOAS on the OTI Research Station Roof. (d) MPL on the beach in front of the Research Station. Map data ©2024 Google.

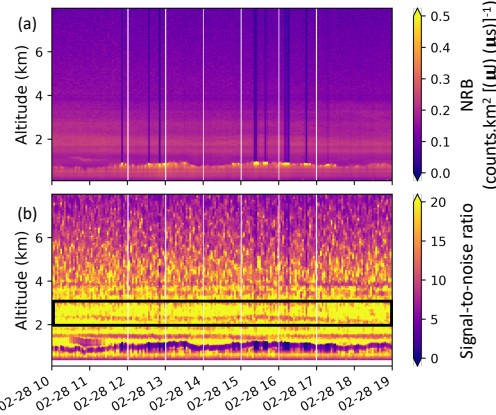

Figure 2: Example MPL backscatter results from 28 February 2023, a mostly clear day, at OTI. (a) NRB between 0-8 km. (b) Signal-to-noise between 0-8 km altitude calculated using moving averages (see text for details). The black box indicates the altitude region with high signal-to-noise ratio identified for calculation of the MPL system constant.

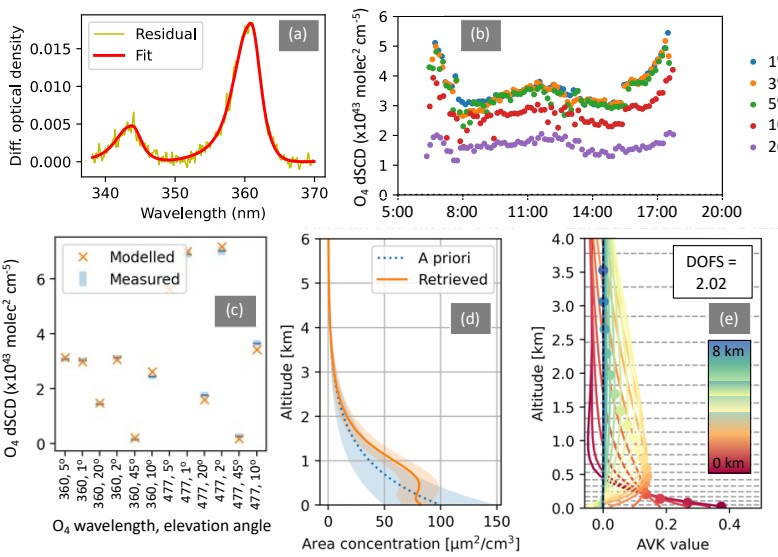

**Figure 3: Components of the MAX-DOAS aerosol retrieval process. (a) Spectral fit for O₄ in the wavelength range 338-370 nm, (b) O₄ DSCD results at 360 nm for a range of elevation angles on 28/2/2023 at OTI (c,d,e) modelled vs measurement DSCD O₄ comparison, a priori and retrieved profile and averaging kernels respectively, from RAPSODI inversion results at 14:20 local time on 28/2/2023.**

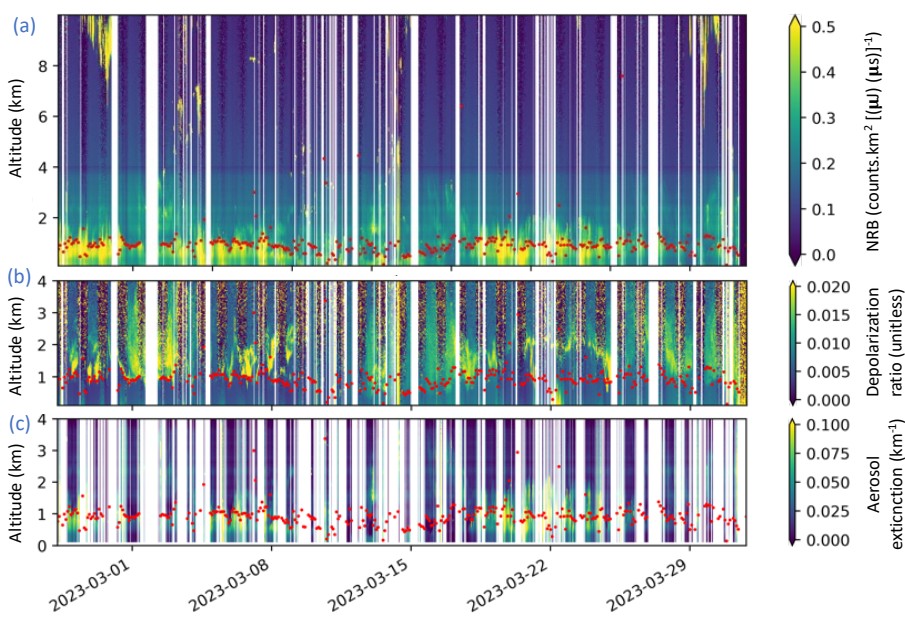

**Figure 4: 30-minute average MPL results during the OTI campaign. (a) Timeseries of vertical NRB profiles from 0-10 km altitude, (b) timeseries of depolarization ratio from 0-4 km altitude and (c) cloud-filtered timeseries of aerosol extinction vertical profiles from 0-4 km altitude. In all plots, red dots indicate the BLH calculated from MPL NRB profiles.**

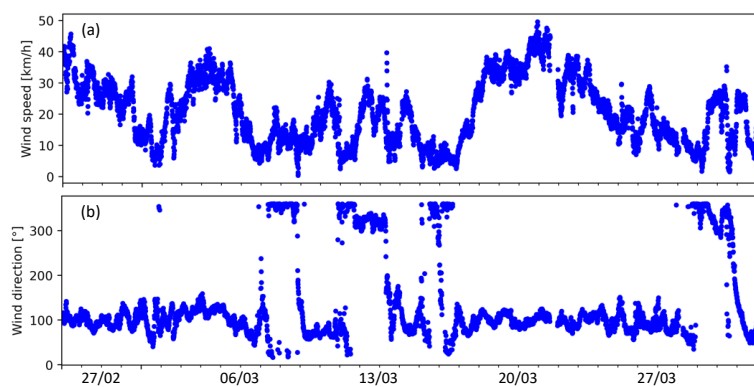

**Figure 5: Wind data from 25 February to 31 March at OTI: (a) wind speed and (b) wind direction.**

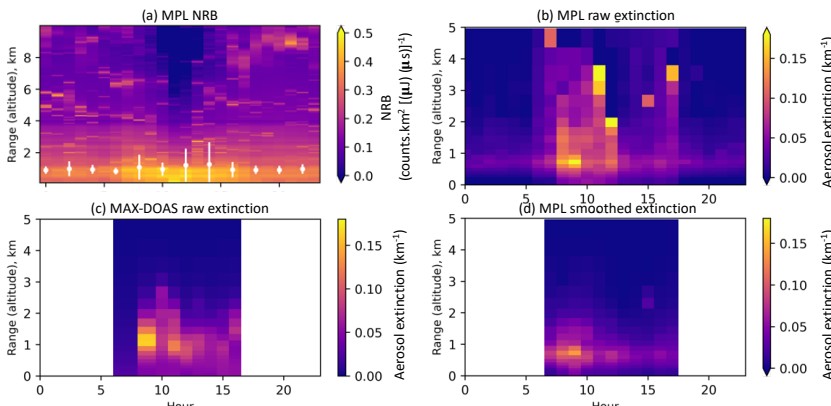

**Figure 6: Diurnal and vertical profile results at OTI. (a) NRB from the MPL (0-10 km altitude), with white dots indicating the mean diurnal variation of the boundary layer height, (b) raw, cloud-filtered aerosol extinction from the MPL (0-5 km altitude), (c) raw, cloud-filtered aerosol extinction from the MAX-DOAS (0-5 km altitude) and (d) aerosol extinction from the MPL, at the MAX-DOAS vertical resolution and smoothed using MAX-DOAS averaging kernels (0-5 km altitude).**

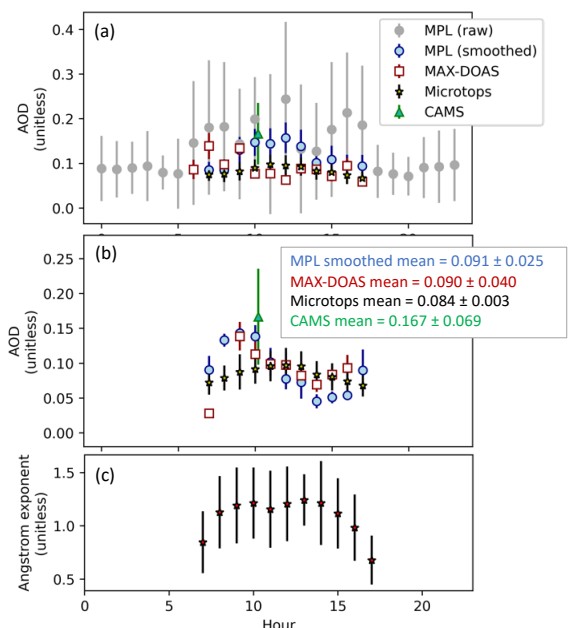

**Figure 7: Diurnal variation of aerosol column parameters at OTI. (a) AOD measured using the three different techniques employed at OTI; Microtops, MAX-DOAS and MPL (both raw AOD and AOD from extinction profiles smoothed using MAX-DOAS averaging kernels). MPL AOD is at 532 nm, Microtops at 500, and MAX-DOAS is adjusted from 360 nm to 500 nm using the Microtops measured Angstrom exponent (see text for details). Also shown is the campaign mean AOD at 550 nm over OTI, for 10 am local time, from archived Copernicus Atmospheric Modelling System forecasts. MAX-DOAS AOD in (a) is from a single wavelength aerosol retrieval at 360 nm. (b) Same as (a) for the MAX-DOAS and smoothed MPL AOD, but using the dual wavelength MAX-DOAS retrieval. (c) Diurnal variation of the Microtops Angstrom exponent (380-500 nm).**

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
