# Peer review of "Measurement report: Aerosol vertical profiling over the Southern Great Barrier Reef using lidar and MAX-DOAS measurements."

_EGUsphere, 2024_

## Author Response (AR1)

**Measurement report: Aerosol vertical profiling over the Southern Great Barrier Reef using lidar and MAX-DOAS measurements**

Robert G. Ryan, Lilani Toms-Hardman, Alexander Smirnov, Daniel Harrison, Robyn Schofield

Submitted to Atmospheric Chemistry and Physics

RESPONSES TO REVIEWERS

Reviewer comments in black normal text. *Responses in blue italic text*.
Note that line numbers referenced herein refer to line numbers in the markup copy of the manuscript.

Reviewer # 1

The manuscript is logically coherent and well-structured. However, the research content appears somewhat simplistic. While related studies have already been published, this research does have the novelty of being the first conducted on One Tree Island in the Southern Great Barrier Reef. Nonetheless, the overall content of the paper may be considered relatively simple for a journal of ACP's standards, and the innovation presented is not particularly strong. I recommend adding new research content to enrich the manuscript and enhance its contribution to the field.

*We thank the reviewer for taking the time to carefully read the manuscript. In response to the comment of simplicity, and request for additional research content, we submit the following:*

1. *The goal of submitting this paper to as a Measurement Report is to provide important, novel measurements and measurement context for atmospheric aerosols and clouds in a previously under-studied location. We submitted this as a Measurement Report to highlight that these measurements exist for model comparison purposes and to inform future measurement campaigns in this area, rather than provide a detailed study of aerosol sources and formation.*
2. *The reviewer notes that this study is novel for being the first from One Tree Island, but to our knowledge it is actually the first overview of aerosol extinction and optical depth measurements in the entire Great Barrier Reef region. The methods used to derive these datasets are far from simple, but demonstrate a sophisticated approach to studying aerosol extinction and optical depth in this remote, challenging environment.*
3. *We also highlight how the interpretation of aerosol extinction and optical depth in this environment is complicated by the choice of method and optical geometry. Far from lacking innovation, this is an important comparison highlighting that multiple techniques and approaches are required to understand the vertical distribution of aerosols using remote sensing methods at the Great Barrier Reef.*
4. *Nevertheless, we have enhanced the novelty and innovation in the study from the original submitted paper, by:*
   - *Re-running the MAX-DOAS aerosol retrieval using two $O_4$ wavelengths, 360 nm and 477 nm. This is the first time this capability has been demonstrated with real-world measurements using the RAPSODI*

*algorithm, significantly enhancing the innovation of the study. The AOD diurnal results in Figure 7 are hence much more consistent between the MAX-DOAS and smoothed MPL retrieval, giving greater confidence in the analysis of the AOD diurnal cycle within the boundary layer. The most important changes in the manuscript resulting from the updated retrieval are:*

- *At Lines 207-208 we now state: "This dimer has strong absorption bands at 360 nm and 477 nm which are used in this work"*
- *At Line 220 we now state: "The wavelength ranges for fitting O4 were 338-370 nm in the UV region and 425-490 nm in the visible region."*
- *Starting at Line 237 we now state: "In this work, the capability of RAPSODI to retrieve aerosol vertical profiles from multiple O4 wavelengths (i.e. 360 nm and 477 nm) simultaneously is demonstrated for with real DSCD measurements in Australia for the first time."*
- *Figure 3 has also been updated to show O4 DSCD retrieval in RAPSODI at 360 and 477 nm.*
- *Figures 6 and 7 have been updated to show AOD results from the dual wavelength retrieval. Note that this has changed not only the MAX-DOAS retrieval but also the MAX-DOAS averaging kernels used to smooth the MPL results.*
- *At several places in the Results, Discussion and Abstract, text has been updated to reflect the new results. In particular, the innovation of new information regarding the appropriateness of the different techniques for different regions of atmospheric aerosol sensing, and the importance of the dual wavelength MAX-DOAS retrieval, is highlighted throughout.*

- *Including further information on analysis of cloud filtering when we compare AOD from different measurement types. Specifically, this involves adapting the cloud camera-facilitated filtering method outlined in Ryan et al., (2024). This information is added to the Methods as follows at line 194: "In Ryan et al, (2024), clouds were identified relevant to the MPL vertical viewing geometry within a $5^o$ radius of the centre of the cloud camera image. For daylight comparison of AOD during the daytime, using the MAX-DOAS and Microtops (which use non-vertical viewing geometries) we also examined the cloud camera images for 'all-sky' cloudiness by using a $50^o$ radius from the centre of the cloud camera image."*
- *This also contributes to the discussion of why AOD values may vary between instruments in the Results, starting at Line 426: "We found no difference in the final AOD diurnal cycles for MAX-DOAS or MPL when considering a wider cloud-filtering threshold from the co-located cloud camera. It made no difference if we applied the cloud filter to only the centre of the cloud camera image, or also invoked a cloud filter from the entire image. This suggests that differences in the AOD amounts between instruments is driven more by differences in vertical structure and sensitivity than horizontal inhomogeneity."*
- *We also provide more information on the filtering of MAX-DOAS results which was missed in the first manuscript version and which adds important information for reproducibility and improves the MPL - MAX-*

*DOAS comparison. Specifically at Line we now state: "Retrievals with poor information content or poor convergence were filtered from the final analysis by rejecting results with less than one degree of freedom for signal and with $\chi^2 < 150$."*

- *We include information on the planetary boundary layer height, calculated using the MPL normalised relative backscatter (see responses to reviewer 2, below, for details and locations of the implementation of this information.*

**Reviewer # 2**

This paper offers valuable aerosol vertical profiles through Max-DOAS and MPL, which should have provided significant scientific insights for the community. However, the current manuscript did not match the journal's quality. Besides, some additional work needs to be addressed. Here are my comments and suggestions:

*We thank the Reviewer for their constructive feedback. We have addressed the concerns as outlined below and believe that this has significantly enhanced the quality of the paper.*

Major comments:

1. line 240-243: "Examining the cloud filtered aerosol extinction profiles in Figure 4(c), we find the high depolarization ratio values directly above the boundary layer are present during both cloud-flagged and cloud-free periods. This indicates a change in the scattering characteristics (shape and/or composition) of aerosols at or just above the boundary layer, whether clouds are present or not." The boundary layer is mentioned several times in the paper, what is the exact altitude of the boundary layer? Is the determination of the boundary layer consistent in MAX-DOAS and MPL?

   - *We have included information on the boundary layer height (BLH) to address this important point. Calculation of the BLH was outlined in Ryan et al., (2024), where a comparison between the MPL BLH and that from drone meteorological measurements was presented. We use the gradient method for calculating the BLH, from the MPL normalised relative backscatter. We now describe this in the Methods section of this paper. We also show the BLH in the revised Figure 4. This enables us to reference the exact value of the BLH throughout the paper. The relevant addition in the Methods (starting at Line 200) is: "Ryan et al, (2024) also calculated the boundary layer (BLH) from MPL observations, demonstrating good agreement between BLH calculated from MPL NRB and drone-measured potential temperature. The BLH provides important context to this study by indicating the likely extent of turbulent mixing in the lowest part of the troposphere. We use the gradient method, as applied in Ryan et al. (2024) to determine the BLH from each MPL NRB profile: this simple method assigns the BLH as the minimum of the gradient of the MPL backscatter profile."*

   - *Figure 4 has been updated to show the BLH time series on all panels, and the caption has also been updated to reflect this.*

- *Starting at Line 293, specifically related to the Reviewer's point about the depolarization ratio discussion, the text has been updated to the following: "The panels of Figure 4 also show the BLH, calculated from the NRB, throughout the campaign. The BLH was typically in the range 0.8-1.2 km throughout (Ryan et al., 2024). In most cases, enhancements in the depolarization ratio were evident immediately above the BLH: the mean depolarization ratio below the BLH was 0.009 ± 0.148, with a high standard deviation due to the occasional presence of clouds, with much larger depolarization ratio, below 1 km."*

- *Starting at Line 306 we also add the following discussion of aerosol extinction and depolarization ratio in relation to the BLH and Figure 4: "Early in the campaign, high aerosol extinction values were typically just below the BLH and below the regions of highest depolarization ratio. Around the 19-23 March, high extinction values were recorded above the calculated BLH, although once again the depolarization ratio was above the region of maximum extinction, as shown in Figure 4."*

2. Equations (10) and (11), as well as a description and explanation of the calculation method, should be placed in section 2.

   - *Equations 10 and 11 and the accompanying explanation/context have been shifted to the Methods as suggested.*

3. Line 333-336: "In almost all hourly bins, the smoothing of MPL extinction by MAX-DOAS averaging kernels brought the MPL AOD closer to the MAX-DOAS AOD. However, the smoothed MPL AOD remains higher than the MAX-DOAS in almost all hourly bins even with vertical sensitivity and resolution accounted for." Can you try to explain why this phenomenon occurs?

   - *We have addressed this concern by testing running the MAX-DOAS retrieval at two wavelengths simultaneously. We describe the relevant changes in more detail in point 4(a) in responding to Reviewer 1, above. In summary pertinent to this point, running the MAX-DOAS retrieval and two wavelengths provides more information to the retrieval which (a) changes the MAX-DOAS mean AOD diurnal cycle and (b) changes the averaging kernels used to smooth the MPL results. This produces a diurnal cycle with much better agreement between the MPL, MAX-DOAS, Microtops and CAMS model.*

   - *The updated paragraph in question now reads as follows (starting at line 454, asterisks mark new text): "Error bars in Figure 7 are the standard deviation of the hourly mean values. The raw MPL AOD has much larger variability than the other techniques. This variability, caused by frequent occurrence of high extinction values, likely indicates that the cloud-filtering algorithm is missing some optically thin cloud above the boundary layer. This results in misinterpreted aerosol information both due to enhanced lidar return and because the extinction-to-scatter ratio calculated for aerosols, RA, will be inappropriate for thin cloud. \*This interpretation is supported by the agreement in magnitude and diurnal cycle shape of the MAX-DOAS and smoothed MPL in the dual wavelength retrieval case. This shows that in a like-for-like comparison in terms of vertical sensitivity, which is only in the lowest km for the MAX-DOAS, both techniques agree well. The raw MPL daytime diurnal cycle is significantly enhanced by being able to observe with higher sensitivity, regions above the boundary*

*layer. The diurnal cycle of smoothed MPL and MAX-DOAS also compares well with the Microtops in the afternoon, although not the early morning, because of more morning cloudy periods.\**"*

- *As a result of these updates to the methods and results, we believe we have addressed the Reviewers' concern about poorly explained AOD agreement, by instead showing better agreement.*

4. Line 349-350: "It is important context for proposed marine cloud brightening experiments at the GBR that aerosol layers can exist up to 2 km altitude." Could you elaborate on the effect of the height of the aerosol layer on marine cloud brightening in the manuscript based on this observation?

- *Thank you for pointing out this lack of detail - we are happy to elaborate on this point. We note that the Introduction already contained extensive reference to marine cloud brightening and the importance of atmospheric aerosol measurements in understanding its potential impacts, e.g. in the paragraph starting at Line 66. In addition, we now draw on the greater confidence from the improved MAX-DOAS results, to have update the paragraph raised here by the Reviewer, and provided greater literature context to the presence of elevated aerosol layers in the GBR region (starting at Line 476):*

  *"The elevated aerosol layers observed here using remote sensing methods are consistent with the in-situ observations in Braga et al. (2025), where elevated aerosol layers above cloud top heights were seen away from clouds and attributed to cloud processing.*

  *The ability to measure aerosol column amounts, layer heights and vertical profiles is also crucial in the context of marine cloud brightening experiments at the GBR. Significant work is already underway to better characterise aerosol amounts and composition in the boundary layer above the GBR using in-situ methods (Braga et al., 2025; Hernandez-Jaramillo et al., 2024). For example, Hernandez-Jaramillo et al. (2024) describe an airborne research facility suited to short-term observation campaigns. While such platforms provide high temporal and spatial resolution within the campaign, a significant advantage of remote sensing platforms such as MAX-DOAS and MPL is that they can be deployed for longer time periods. The agreement in AOD amounts and diurnal cycles between MAX-DOAS (dual wavelength retrieval) and MPL shows that both techniques are suitable for probing boundary layer aerosol distributions. We therefore suggest that deployment of one or both techniques to a location like One Tree Island, over a long time period, would be advantageous to determine long-term and season boundary layer aerosol variations at the GBR. However, our results highlight two important points about deploying these different sensors for long-term applications. Firstly, the low sensitivity of MAX-DOAS above the boundary layer means that technique misses a significant proportion of the aerosol extinction column in this environment. Secondly, running the MAX-DOAS retrieval at 360 and 477 nm simultaneously is a more robust method of examining boundary layer aerosol extinction and optical depth, compared to at 360 nm only."*

5. The aerosol vertical profile and the aerosol optical depth are analyzed in detail from MAX-DOAS and MPL at One Tree Island in the Southern Great Barrier Reef in the paper. In this observation, can you provide a comparison of the similarities and differences between the results of the two measurement and their respective applications?

- *We address the importance of measuring aerosol vertical profiles in the GBR region several times throughout the paper, including in the Introduction and discussion (see also the response to previous comment), and in the Results when talking about elevated aerosol layers observed in the raw MPL profile and also regarding elevated layers of high depolarization ratio. To place the AOD in its own context, we amend the AOD Discussion paragraph to the following (starting at Line 510):*

  *"The importance of examining AOD is highlighted by the range of possible AOD sources. AOD is a metric for considering aerosol/radiation metrics available from satellite platforms, ground-based remote sensing and modelling frameworks (e.g. CAMS), in cases where vertically-resolved measurements are not available. Therefore, building trust in available AOD measurements is crucial. The comparison of AOD measurements between instruments at One Tree Island found mean values of 0.084 ± 0.003 for the Microtops, 0.090 ± 0.040 for the MAX-DOAS (dual wavelength retrieval) and 0.091 ± 0.025 for the MPL (smoothed for comparison with the MAX-DOAS). Good diurnal cycle agreement was found between the MAX-DOAS and smoothed MPL diurnal cycle and also with the afternoon part of the Microtops diurnal variation. The morning value for these instruments also agreed within uncertainty with the forecast AOD from the Copernicus Atmospheric Modelling System, a widely accessed source for public information on AOD, providing confidence in our understanding of the daily variability in boundary layer AOD levels. In future work it would be good to examine the reasons for the morning peak in lower tropospheric aerosol extinction and optical depth."*

Minor comments:

1. Line 233: "0.112 and 0.294 counts.km2 [(**µ**J) (**µ**s)]-1", "**µ**" should be "µ". Same as Figure 2

   - *Fixed*

2. Line 238: "pattern of being <0.01" needs space between "<" and "0.01".  Same as line 238: " >0.02", and line 325: " a factor >2.5".

   - *Spaces have been added*

3. In Figure 3 Pictures (a) and (b) are resized consistently, and (c) (d), and (e) will look better if they are resized consistently. In addition, the background color of (a), (b), (c), (d), and (e) can be removed.

   - *We appreciate the reviewers' thoughts here, however, given the differences in each panel and the large number of colours, different styles (lines, dots etc), we would prefer to keep the sizes and labels of Figure 3 as they are.*

---

## Author Response (AR2)

Ryan et al., (2025), Submitted to ACP:
*Measurement report: Aerosol vertical profiling over the Southern Great Barrier Reef using lidar and MAX-DOAS measurements*

**Responses to the Editor, following addressing Reviewer Comments.**

Editor comments in black text *- Author responses in italic blue text*

*We thank the reviewer for their support of our manuscript, pending fixing the minor revisions requested. We have addressed the comments, with the exception of struggling to know how to address the formatting issues related to parentheses alignment. Will it be possible to submit the manuscript as a Word document for copy editing, rather than as a PDF? The equations, subscripts and parentheses alignment look good in Word.*

1) Please remove the period from the title. *- fixed*

2) Abtract: RAPSODI is not defined. *- fixed*

3) Line 85: the time of experiments is January to March, 2023. In the abstract the time is from February to April, 2023. Please clarify. *– Clarified in the methods at lines 85-86*

4) Last paragraph of the introduction: Please reformulate the paragraph in a form that provides the aim of this paper. At least the first sentence needs to be rephrased. *– paper aims reformulated in lines 75-79.*

5) Line 98: please check the observation period to be consistent. *– observation periods fixed*

6) Line 131: GPS is not defined. *- fixed*

7) Equation 3-6, please correct the alignment of the parenthesis in general and around (r) as well as throughout the text. *– thanks for pointing this out; the alignment of the parentheses are correct in my Microsoft Word version, however I can see that the alignment is wrong when converted to PDF format. My understanding is that in the final submitted paper, copy editing of equations will be updated by copy edit staff – I am not sure what else I can do at this stage because it is fine in Word.*

8) Line 192, please define One Tree Island (OTI) in the text. *- fixed*

9) Line 218, please define QDOAS algorithm. *– Differential optical absorption spectroscopy is already defined, and QDOAS is just the name of the algorithm rather than a further acronym. I have addressed this comment by putting quote marks around QDOAS the first time it appears.*

10) Equation 8 and text around it. Please correct the alignment of the parethesis. *– see response to point 7 above.*

11) Please use proper subscript for $O_4$ instead of just smaller font. *– I have done this throughout already. Once again I think this is an issue with the conversion from Microsoft Word to PDF, or perhaps my choice of Times New Roman font.*

12) Line 234 and elsewhere (figures etc), please use a consistently a fixed format for the dates. *- fixed*

13) Equation 9 and related discussion in the text, please fix alignment of the

parenthesis. *– see response to point 7 above.*

14) Line 249, please define VLIDORT algorithm. *- fixed*

15) Line 273, please rename the section to Results and Discussion. Please consider splitting the section into at least 2 separate sub-sections. *- fixed*

16) Line 278, please indicate, where the full data set is available. *– the full data set is available as indicated in the Data Availability section at the end of the manuscript.*

17) Line 286, please remove the period before the unit. *- fixed*

18) Line 296, please indicate the mean depolarization rate and the standard deviation in the cloud-free conditions as well. *– this information has been added at line 287.*

19) Line 305, The highest... *- fixed*

20) Paragraph starting from line 305, please use consistent way to express the date and include the year. *- fixed*

21) Line 332, the greatest *- fixed*

22) Line 395, why do you need to present the equation here? – it is important to mention the Angstrom exponent equation for reproducibility *– anybody else wanting to analyse or compare AOD at different wavelengths using different techniques will find this useful.*

23) Line 433, please remind the reader, what is the value of AOD based on your results. *-fixed*

24) Line 439, is there a way to remove the cloud impact? *– as described in the Methods, we go to some considerable extent to filter out clouds. The point made at line 439 is that despite this, the variability in raw MPL-derived AOD is still high. My hypothesis is that high level cloud is being missed and misinterpreted as AOD. I have rephrased this section to the following "The raw MPL AOD has much larger variability than the other techniques. This variability, caused by frequent occurrence of high extinction values, may indicate that the cloud-filtering algorithm is missing some optically thin cloud above the boundary layer. This could be tested with cloud filters based on different cloud detection methods, for example different LiDAR types not employed at One Tree Island. Inappropriately flagged cloud would result in misinterpreted aerosol information both due to enhanced lidar return and because the extinction-to-scatter ratio calculated for aerosols, RA , would be inappropriate for thin cloud."*

25) Line 447, please change the section name to Conclusions and have the discussion in the section "Results and Discussion" concentrating on the Conclusions here. *– fixed, especially by shifting the last part of the previous 'Discussion and conclusions' section into the end of the updated 'Results and Discussion' section.*

26) Line 494, please rephrase AOD source. There are sources that increase aerosol concentrations that influence AOD. *- fixed*

27) Line 500, CAMS *- fixed*

28) Figure 2. Please be consistent with the date format. *- fixed*

29) Figure 3. DOFS is not defined. *– fixed in the caption*

30) Figure 4. Please be consistent with the date format. *- fixed*

31) Figure 5. Please be consistent with the date format and please include the year. *- fixed*